# Virtual lab coats: The effects of verified source information on social media post credibility

**Jorrit Geels**[1,2]*, **Paul Graßl**[1], **Hanna Schraffenberger**[1,2], **Martin Tanis**[3], **Mariska Kleemans**[4]

**1** Interdisciplinary Hub on Digitisation and Society, Radboud University, Nijmegen, The Netherlands,
**2** Institute of Computing and Information Sciences, Radboud University, Nijmegen, The Netherlands,
**3** Department of Communication Science, Vrije Universiteit, Amsterdam, The Netherlands, **4** Behavioural
Science Institute, Radboud University, Nijmegen, The Netherlands

* jorrit.geels@ru.nl

**Data Availability Statement:** All relevant data are within the manuscript and its Supporting information files.

**Funding:** The author(s) received no specific funding for this work.

## Abstract

Social media platform's lack of control over its content made way to the fundamental problem of misinformation. As users struggle with determining the truth, social media platforms should strive to empower users to make more accurate credibility judgements. A good starting point is a more accurate perception of the credibility of the message's source. Two preregistered online experiments ($N = 525$; $N = 590$) were conducted to investigate how verified source information affects perceptions of Tweets (study 1) and generic social media posts (study 2). In both studies, participants reviewed posts by an unknown author and rated source and message credibility, as well as likelihood of sharing. Posts varied by the information provided about the account holder: (1) none, (2) the popular method of verified source identity, or (3) verified credential of the account holder (e.g., employer, role), a novel approach. The credential was either relevant to the content of the post or not. Study 1 presented the credential as a badge, whereas study 2 included the credential as both a badge and a signature. During an initial intuitive response, the effects of these cues were generally unpredictable. Yet, after explanation how to interpret the different source cues, two prevalent reasoning errors surfaced. First, participants conflated source authenticity and message credibility. Second, messages from sources with a verified credential were perceived as more credible, regardless of whether this credential was context relevant (i.e., virtual lab coat effect). These reasoning errors are particularly concerning in the context of misinformation. In sum, credential verification as tested in this paper seems ineffective in empowering users to make more accurate credibility judgements. Yet, future research could investigate alternative implementations of this promising technology.

## Introduction

In our digital- & information-society, misinformation has become a fundamental problem. A recent example includes a wave of misinformation in November 2022, right after Twitter (now 'X') announced it paid subscription [1]. The new policy included that every user could pay $8

**Competing interests:** The authors have declared that no competing interests exist.

to get a blue verification badge, implying their identity was verified. However, users abused this badge to imply a verified identity with the goal of impersonating companies and spread misinformation. Many readers believed the accounts belonged to the impersonated companies and accepted their misinformation posts as true. Consequently, the stocks of the impersonated companies dropped [1].

Events like these illustrate the relevance of reliable source information when users look for credible information online. Source information is typically provided by verifying the *identity* of account holders. For company accounts, certainty over their identity is often sufficient to prevent impersonation and misinformation. Yet, for (unknown) people, certainty over their *identity* provides little extra information on the credibility of their information. It can therefore be dangerous if users conflate verified source identity and message credibility.

For more accurate content evaluation, certainty over their *credentials* (e.g., employer, role) can be desired instead [2], as they could signal expertise. For example, if an unknown person makes a medical claim and one were to doubt its credibility, it is much more valuable to know their medical credentials rather than their identity. Therefore, this paper compares the behavioural effects of source *identity verification* and the novel approach of source *credential verification* to reduce the impact of misinformation in social media.

## The birth of modern misinformation

It is incredibly important that internet users can easily find accurate information. In the traditional media landscape (e.g., newspapers), information is typically checked on its truthfulness by so-called gatekeepers (e.g., journalists) prior to publishing [3]. However, people increasingly use social media as their primary source of news [4]. Because social media's nature, anyone can say something, and traditional institutions have less authority [5]. However, this could come at the expense of information quality because not everyone has the same expertise [6]. For example, social media has become an important source of medical information [7, 8]. Especially in that case, it is crucial that people rely primarily on expert advice.

Given the reduced authority of the traditional gatekeepers on social media, the responsibility of checking for truthfulness is shifted away from these gatekeepers to the end-users [9, 10]. However, accuracy assessments can be difficult and time-consuming for users [10]. Hence, they often use simple rules of thumb (i.e., heuristics) to quickly make accuracy assessments and decisions regarding whether to share certain information [11–13]. When such quick assessments are inaccurate, it can have disastrous consequences. For example, take the prominent role of online misinformation in the COVID-19 pandemic (e.g., [14, 15]), the US Capitol riot (e.g., [16]), and the war in Ukraine (e.g., [17]).

To help people make these decisions more accurately yet still quickly, a popular approach is fact-checking. However, research has shown various problems with fact-checking. First, despite many social platforms using tags such as 'disputed' or 'false' [18], the effect of such tags is limited [19–22]. Moreover, manually tagging all social media posts is infeasible [23–25], and automated detection cannot fully replace this process [26]. Next, when only tagging a subset of social media posts, unlabelled posts could then be perceived as accurate even if they are not [27]. Last but certainly not least, countering inaccurate beliefs through debunks might not be effective (e.g., [28]), if such debunks are even noticed at all. It thus seems the user cannot be completely excluded from credibility assessment.

Instead of a platform-centred approach to reducing the impact of misinformation using truth labels or debunking information, another (perhaps more promising) approach is to empower the end-user themselves. Examples of this alternative user-centred approach often include media literacy education [29], to help users improve the accuracy of their judgements.

Yet, media literacy education might not be accessible to all [24], and there is also evidence suggesting that media literacy education by itself does not necessarily influence the impact of misinformation [29–31]. Hence, users should be empowered through the platform itself [24, 32].

## Empowering users for more accurate truth assessment

A social media platform cannot fully control its content, but it can instead make its users smarter about which content they believe to be true. A user's accuracy of determining truths is shaped by how credible the content appears to be to a user [33]. For online content, these accuracy assessments are known to depend on four types of factors, namely the characteristics of the end-user, platform cues, message cues, and source cues [34]. Characteristics of the end-user that can affect determining credibility include factors such as their prior knowledge, frequency and purpose of internet usage, as well as demographic variables like age. Platform cues, such as the aesthetics of the website or the presence of certificates or recommendations from trusted sources, can also impact credibility perception. Next, message cues like the presence of a timestamp, the use of professional and clear writing, or the plausibility of the message itself can influence its perceived credibility [34].

Lastly, the credibility of content is heavily evaluated through the lens of perceived source credibility [35]. Moreover, source credibility is arguably the feature that yields the most accurate message credibility perception [36]. Source cues that influence credibility judgements include factors such as the degree to which a user identifies with the author, author qualifications and credentials signalling their expertise, or their reputation [34]. For example, knowing the origin of a message, such as the newspaper or journalist behind it, can greatly impact how users perceive its credibility. The underlying assumption here is that statements from credible sources are considered more trustworthy than those from less reliable sources.

Therefore, providing information about the source can help users make more accurate source and message credibility judgements. Unfortunately, social media platforms often provide limited information about a source and its credibility. Still, source credibility cues yield a useful heuristic against misinformation [37, 38]. With the goal of empowering users to make more accurate credibility judgements of online information, this paper aims to explore how various cues of source credibility on social media influence users' evaluations of credibility.

## Verified source information

Most social media platforms, to some degree, already provide source cues in the form of verification badges. Namely, accounts of public interest (e.g., celebrities, public institutions, governments, etc.) can obtain a verification badge: the social media platform has then verified that the holders of these accounts are indeed who they claim to be [39, 40], i.e., *authentic*. Current developments on Twitter, Instagram and Facebook include paid subscriptions to obtain verification badges [1, 41]. Introducing verification badges as a subscription makes these badges slightly more accessible to the public. However, there is reason to believe that users might conflate source authenticity (i.e., verified source identity) and message credibility [42, 43]. The literature is unclear on the effects of source authenticity on source and message credibility, partly due to its methodology (see Related work for more details). This paper therefore re-explores the effects of source authenticity badges on social media.

Moreover, this study explores potential new solutions to improve source credibility judgements on social media. As posed before, it seems more useful to have certainty about the source's credentials rather than their identity when judging information. To elaborate, a verification badge confirms that the unknown sources are *who* they claim to be, i.e., its authenticity. Yet, it often is more relevant to be sure they are *what* they claim to be (i.e., in terms of

credentials or expertise). For example, it might be helpful to know whether a post about COVID-19 comes from a medical professional, whether someone predicting earthquakes is a geologist, or whether someone describing political unrest is located in the city they are writing about.

Such (verified) credentials can be a useful heuristic for assessing credibility [44], and potentially mitigate the influence of misinformation on an individual's beliefs and their inclination to share the information [45]. However, credentials of an unknown author are currently commonly displayed (if available) in their account's biography, which is rarely used in credibility judgements as it is not directly accessible [37]. Even when such information is disclosed in the biography, it is self-reported, and users are unlikely verify this information [13, 46], possibly because it requires more effort [46]. Still, when visible at a glance, *credential verification* could empower the user in evaluating the accuracy of content, while reducing the required effort to do so [47]. Lastly, credential verification could make social media more accessible, as it could give a voice to those whose expertise is not well-known, i.e., evident from their identity being verified.

An interesting development here is the introduction of digital identity wallets [48], which are mobile applications that enable users to collect, store, and share verified (identifying or non-identifying) personal data. Examples of such data are city of residence, nationality, educational degrees, or certificates [49]. Thus, credential verification could prove an unknown author's credential and be displayed together with their social media post [2], following the suggestion that source information should be visible at a glance [36, 37].

However, providing such verified information comes with technical, ethical, and societal challenges. The technical feasibility of a credential verification system on Twitter has been demonstrated [2]. Still, showing verified attributes of a person comes at the cost of privacy, could emphasise more *verifiable* data (e.g., degrees) over less tangible expertise (e.g., self-taught skills), and could potentially be misinterpreted.

Overall, *credential verification* sounds like a promising heuristic. However, we fear that the credibility of a source with verified credentials might be inflated when discussing topics outside their domain of expertise (henceforth the "virtual lab coat effect"). Therefore, before developing an infrastructure that provides verified credentials, it is crucial to evaluate its potential impact on users. Though user perceptions and potential misconceptions have been explored in pilot studies [50, 51], this paper presents the first two large-scale empirical studies on identity and credential verification of unknown sources on social media. Intermediate results are discussed per study, after which the paper concludes with a general discussion.

## Related work

Metzger et al. [52] conducted research on how users make reliability judgements online. They argue that credibility judgements of information from traditional media may not necessarily be formed in the same way as judgements of information from online media. Their study identified five heuristics that users employ to assess the credibility of online information. They show that the *reputation* of a source is an important heuristic for evaluating the credibility of online content. Here, the credibility judgement is primarily based on prior interaction with the source and its familiarity. Next, using the *expectancy violation* heuristic, users consider whether their expectations are met by the content (e.g., in professionalism). The other three heuristics users tend to employ are *endorsement* (i.e., credibility judgements based on the assessments of others), *consistency* (i.e., cross-checking content with different sources), and *persuasive intent* (i.e., suspicion of hidden agendas, such as with commercial content).

However, the findings by Metzger et al. [52] do not exclude the possibility of users considering additional information when it is provided to them. Namely, users are likely to incorporate all available information to form credibility judgements about the source [53]. Therefore, it is interesting to examine additional cues, such as verified identity or credentials, and explore their effects on source and message credibility.

Furthermore, many studies have investigated sharing behaviour of users in the context of misinformation (see, e.g., [29, 54–62]). An important notion in this domain is that credibility and sharing judgements have recently been found to be two separate, not necessarily related decisions (see, e.g., [29, 56, 62]). For example, users might share information they know to be inaccurate but would be 'interesting if it were true' [59]. Moreover, misinformation is often shared without sharers being aware of its inaccuracy [61, 62]. Hence, credibility judgements and sharing intentions should be considered independently.

The relationship between source credibility and sharing has been explored in a similar research by Kim & Dennis [12]. In their experiment, some posts included source credibility ratings in the form of, e.g., star ratings. They found that when including such credibility ratings, users became more critical of all incoming information and decreased in sharing behaviour. These results illustrate how the design of social media posts can affect both credibility and sharing behaviours. The authors call for investigations on how author information influences credibility and sharing behaviour. The present paper investigates the effects of two types of (verified) author information: identity verification, and credential verification. Related work on either approach is discussed below.

## Identity verification

A handful of researchers have investigated the effects of verified source identity on the credibility of their message. There namely is reason to suspect that users conflate source authenticity and message credibility, implying that messages from authors whose identity is verified are more credible. For example, there is evidence to suggest that claiming to be an 'official' (i.e., authentic) source positively affects ones perceived message credibility [63]. However, that study used websites as stimuli, so the results do not necessarily generalise to social media accounts. Next, participants in the experiment of Morris et al. [37] explicitly mentioned that verification of a source's identity positively affected the credibility of the message. Yet, Morris et al. do not provide behavioural evidence. It thus remains unclear whether their findings represent analytical thinking or heuristic thinking, which is an important distinction as the latter dominates credibility judgements on social media [52].

In contrast, some scholars did not detect a relationship between account verification and message credibility [39, 42]. This suggests users may simply not conflate source authenticity and message credibility. For example, Vaidya et al. state "users generally understand the meaning of verified accounts" [42] (p11). Still, some caveats in the methodology of [39, 42] are to be noted. First, Vaidya et al. [42] measure message credibility as the degree to which one is likely to adopt or share the message. However, as mentioned, credibility and sharing judgements are not necessarily related. Alternatively, their results can possibly explained by that only few people tend to often re-share information (e.g., [56, 58]). Hence, separately measuring message credibility instead (e.g., through the scale by Appelman & Sundar [64]) reveals more insightful information about whether users conflate source authenticity and message credibility. Moreover, research on the relationship between account verification and message credibility concludes there is no effect based on the absence of evidence of an effect [39, 42]. However, absence of evidence does not mean evidence of absence [65]. The analysis methods used in their studies cannot detect absence of an effect (e.g., [66]).

In sum, the effects of identity verification badges on perceived source and message credibility are unclear. Still, social media platforms like Twitter and Meta employ verified identity badges [1, 40], but also signed the Code of Practice, thus promised to aid users in identifying trustworthy content [32]. If a verified identity makes any content more credible, these badges may even boost misinformation, especially if they are more accessible to the public. Therefore, the effects of identity verification on source and message credibility as well as sharing behaviour should be explored more in-depth using methods that can detect whether an effect is present, e.g., Bayesian models (e.g., [66]). Namely, in contrast to classical frequentist statistical methods, Bayesian models can both reject as well as *accept* null hypotheses.

## Credential verification

Still, theoretically, it is sound that learning the identity of some unknown source is verified does not affect credibility of their message. A potentially more useful heuristic to approach message credibility, is source credentials [36]. Namely, users perceive messages from experts as more credible, but this effect only seems to occur when the user is uncertain about the information accuracy [44], e.g., when the topic is not in the domain of expertise of the user. For instance, scholars [67–69] found that online health information from a source with medical expertise is perceived as more credible compared to a source without clear expertise. Lee & Sundar [70] found that when a source has many online followers, their content is perceived as more credible when the source claims expertise. Yet, when an author has a small audience, information from an expert is perceived as less credible compared to an author without claimed expertise.

In terms of the heuristics identified by Metzger et al. [52], verified credentials align with the *reputation* heuristic. For instance, although the specific medical professional may not be familiar, the general reputation of medical professionals is. Verified credentials are thus likely more informative than verified identity for unknown sources, as verifying the identity of a source reveals nothing about their credentials (i.e., reputation), an important heuristic [52]. Yet, the reputation heuristic poses the danger of a "virtual lab coat effect" (i.e., verified expertise yielding increased credibility on topics outside of domain expertise), as users might not consider whether the reputation is relevant to the message. For example, evidence from online review literature suggests that every extra piece of information about a source improves the credibility of their review [71–73].

However, the *expectancy violation* heuristic could be interpreted more broadly to serve as a counterargument against the occurrence of such a virtual lab coat effect. Namely, if some medical professional provides information on a different topic, this may deviate from the user's expectations and therefore be perceived as less credible. In sum, it is unclear whether verifying source credentials yields a virtual lab coat effect. Therefore, it is crucial to carefully examine the effects of verified source credentials on social media.

## Study 1

The goal of this first study was to explore the effect of source credibility cues on information of unknown sources on social media, contributing to research on countering misinformation. The misinformation problem is fuelled by people believing and/or sharing misinformation. Hence, it is investigated how source credibility cues affect perceived source and message credibility, as well as sharing intentions. Note that this paper refers to Twitter rather than 'X', as we conducted our study before X was launched. Specifically, study 1 was conducted in August 2021, i.e., before the big changes on Twitter [1].

To enable easier comparison of our results, we adopted the stimuli and hypotheses 1a/b/c based on the work by Vaidya et al. [42]. Next, study 1 used a similar methodology, thus focusing on Twitter. Compared to Vaidya et al. [42], the present study differs in the conceptualisation of credibility. Namely, we separately measured source credibility, message credibility, and sharing intentions. As noted above, previous work in this area used frequentist statistical methods, which, in contrast to Bayesian methods, cannot yield conclusions on the presence or absence of an effect (e.g., [66]). Hence, the data were analysed with Bayesian models to determine the presence or absence of the relationship between source credibility cues and the aforementioned measures. Another important reason to conduct a Bayesian analysis instead of a frequentist analysis, is because Bayesian models often align more closely with how humans conceptualise and interpret parameter estimates [74].

We conducted an online experiment to investigate the effects of two types of source credibility cues on social media. First, we investigated the effects of the commonly used identity verification badges, formulating H1a/b/c to confirm the results by Vaidya et al. [42]. Namely, the authors report that identity verification badges have limited to no effect on perceived credibility or sharing on Twitter. Second, we investigated the effects of new envisioned credential-badges, containing verified employer information about social media sources. Such extra information should help users make more accurate judgements, thus only affect the user's judgement if the verified credential is relevant to the context. Accordingly, this led to the following hypotheses:

- **Hypotheses 1a/b/c**: People find Tweets with an identity verification badge equally credible (a: source credibility, b: message credibility), and are equally inclined to share them (c: likelihood of sharing), compared to those without a badge.

- **Hypotheses 2a/b/c**: People find Tweets with a credential verification badge more credible (a: source credibility, b: message credibility), and are more inclined to share them (c: likelihood of sharing), compared to those without a badge, only if the credential is relevant to the context.

Moreover, this study focuses on how users intuitively form credibility judgements and sharing intentions based on looking at the Tweets. First, this aligns with the heuristic evaluation commonly employed on social media [52]. Second, to the best of our knowledge, social media on-boarding does not include information about how to interpret identity badges. Hence, not including information on the interpretation of these badges prior to exposure seems most ecologically valid.

Still, the study aimed to empower users to more accurately assess online information. Therefore, the experiment consisted of two rounds: a first round with *intuitive responses*, and a second round with *informed responses* as part of an exploratory analysis. Namely, prior to this second round, users were provided with information about the interpretation of the identity and credential verification badges. This setup loosely followed the advice of Fallis [33] that users should receive instructions of what features are indicative of information accuracy.

## Method study 1

Before running the experiment, we preregistered the sample size estimation, hypotheses, and planned statistical analysis. The preregistration and all used stimuli, data, and analysis scripts are available on the Open Science Framework (https://osf.io/cyh8e/).

Study 1 was approved by the Ethics Committee and Faculty Board of Social Sciences at Radboud University. The ethical approval used reference number ECSW-2021–079. As part of the

ethical research procedure, participants expressed informed consent through an electronic form prior to the experiment.

**Design.** The experiment used a 3x2 between-subjects design, with each participant viewing one Tweet. There were two independent variables: First, the source credibility cue, with the three conditions: no badge (control condition), identity verification badge, and credential badge which confirmed the author works for University College Hospital. Second, the Tweet *context*, with two conditions: a *health*-related context where the medical profession is relevant and a *non-health* (household) context where the medical employment is not relevant. There were three dependent variables: First, sharing likelihood, indicating how likely a participant is to share the information in the Tweet with others. Second, source credibility, indicating how much credibility a participant attributes to the Twitter account. Third, message credibility, indicating how much credibility a participant attributes to the Twitter post.

**Stimuli and setup.** A (fictional) Tweet was created for each of the six conditions, where each of the six conditions was a combination of one of three badge types and one of two contexts. The medical Tweets are displayed in S1 Fig (all stimuli are available on Open Science Framework). The experiment was hosted on a licensed LimeSurvey server from the host university, where the Tweets were displayed as screenshots.

For the experiment, a realistic yet generic profile picture and name was chosen. All factors of the Tweet apart from the source credibility badge were kept constant across conditions. The number of retweets (101) and likes (260) as well as Tweet contents were taken from [42] to enable more direct comparison of results. The text for the medical Tweet therefore read "There is an increased risk of hypertrophic cardiomyopathy in people who drink 4+ cups of coffee per day." Next, the non-medical, household-related Tweet read "This year's increase in cattle disease will cause 12% jump in average household's grocery bill".

**Participants.** On 30 August 2021, we recruited a total of $N$ = 525 participants, based on a G*power analysis. Note that this analysis is typically only used for frequentist statistic analyses, as opposed to Bayesian analyses. Still, the G*power analysis was purposely used to indicate an 'ethical' sample size and to enable other researchers to re-use our data for frequentist analyses.

Our sample was recruited through Prolific Academic, recruiting anyone with an age between 18 and 65 years (to represent a broad range of society) and their current living location in the United Kingdom (to minimise noise in the data because of cultural differences). Participants were compensated with £1 for successfully completing the study, which was estimated to take around 8 minutes (£7.50/h). On average, it took participants 4.13 minutes ($SD$ = 2.68) to complete the study. The total sample population had a mean age of 31.36 years ($SD$ = 11.11). A pilot study with 20 people was conducted prior to the main data collection to confirm whether everything was clear for the participants and worked as expected.

**Procedure.** Before the experiment, we presented participants with information about the study. The topic only indicated 'Perception of Twitter posts' as a topic, intentionally not touching upon credibility to circumvent possible bias. After the informed consent procedure, we asked participants demographic questions and background questions about their Twitter use, trust in medical organisations, and media literacy. Subsequently, participants were randomly assigned to one of the six conditions and thus presented one of the six Twitter posts. They were asked to look at the Tweet and read its contents. Then, they were asked to report on their likelihood of sharing and perceived source and message credibility. In that order, to not bias the sharing decision by the explicit credibility questions. The questions were displayed on a separate page from the Tweet, to assess user's heuristic evaluation rather than their critical evaluation. Next, the survey included several questions to check if they had paid attention to the Tweet and could recall whether—if any—account badge they had seen.

After this 'first round', it was explained to the participants how to interpret the different possible badges included in the experiment. Here, it was explained the Tweet's author could either have no badge, an identity badge, or a credential badge. The information also included explanation on how to interpret the two badges, namely either as verified identity or as verified credentials respectively. The full explanation text can be found in S1 File.

This information was followed up by the Tweet they had already seen and were asked to complete the same credibility and sharing questions as before, to explore the effects of this information. In the end, we thanked the participants for taking part in the experiment and debriefed them about the purpose of this study. The debriefing made clear that all shown Tweets and their content were purely fictional.

**Measures.** Prior to viewing the Tweet, the participants answered a few background questions. First, Twitter use was assessed on a four-point scale ranging from *Never* to *Often* (higher values indicate more Twitter use; $M = 3.39$, $SD = 0.73$). Next, it was measured how much participants trust employees of medical organisations, by asking "How much of the time do you think you can trust medical organisations to do what is right?". To answer, a four-point scale ranged from *Never* (1) to *Always* (4), $M = 3.05$, $SD = 0.58$. Lastly, media literacy skills were assessed using the validated scale from Vraga et al. [75]. This scale is comprised of six items on a seven-point scale ranging from *Strongly disagree* (1) to *Strongly agree* (7) [75], $M = 5.51$, $SD = 0.84$, $\alpha = .78$. Example items are "I have the skills to interpret news messages" and "I'm often confused about the quality of news and information".

Three dependent variables were measured: sharing intentions, message credibility, and source credibility. We used a five-point Likert-scale ranging from *Very unlikely* (1) to *Very likely* (5) from Vaidya et al. [42], to measure the likelihood that participants would share the information provided in the Tweet. Though self-reported sharing intentions conceptually differs from actual social media sharing behaviour, they are correlated [76]. Although sharing is a binary decision in practice, wider scales are used to measure this concept more sensitively.

Next, to measure how credible participants found the Twitter account holder, we used the five-point scale from Metzger et al. [77]. Participants had to report how biased, trustworthy, professional, and credible they found the author, with the answer options ranging from *Not at all* to *Extremely* (higher values indicate more credibility). The source credibility measure showed good internal consistency with a raw Cronbach's $\alpha = .80$. Dropping the item 'biased' would increase the overall $\alpha$ by .11. However, this is not advisable because the scale only consists of few questions, and internal consistency is good even without modifications.

To assess message credibility, participants rated how accurate, believable and authentic they found the content of the Tweet through a seven-point Likert-scale, ranging from *Strongly disagree* to *Strongly agree* (based on [64]), $\alpha = .90$.

Lastly, we included some checks to assess whether participants paid sufficient attention. First, 8 participants were excluded from the analysis as they could not remember the content of the Tweet. Next, participant also were excluded from analysis if they thought they knew the fictional account, to prevent familiarity bias (only 1 out of 533 participants). Finally, participants were asked to report which badge the author of the Tweet had, where 40.5% reported they cannot remember, 38.5% remembered correctly, 21% did not remember correctly. Clearly, many participants guessed, supporting that source credibility cues are used as a heuristic rather than input for elaborate reasoning. Still, participants were not excluded from analysis based on their answer, as we are interested in both heuristic and more informed effects of source credibility cues. An exploratory analysis performed on the sub-sample that only considered the participants that answers the recall question correctly produced the same general pattern of results compared to the main analysis.

**Analysis.** In the main analysis, we measured the main and interaction effects of source credibility cue and context on sharing intentions, message credibility, and source credibility. The model controlled for the participants' Twitter use and media literacy. Trust in medical organisations was excluded for robustness and reliability, i.e., because this factor was not measured using a validated scale. Note that controlling for this variable yielded mostly similar outcomes to the main analysis.

The main analysis used the values of the participants' *intuitive response*, i.e., the data obtained after participants had seen the Tweet for the first time, without any prior explanations about the meaning of the different badges. To explore the effects of an information intervention on verified source information, the main analysis also included the data of the participants' *informed response*. This data was obtained after the meaning of the badges had been explained, and participants had seen the Tweet for the second time. For both responses, none of the results were heavily influenced by outliers or participants' age. Note that both the intuitive and informed response were part of the same model, to circumvent an inflation of Type-1 error results.

The analysis was done using a Bayesian multivariate cumulative model to fit Likert-scale data [78], applying a logit link function. The multivariate model allowed us to consider all outcome levels and their relationship with each other in the same model. Instead of classic significance testing, we used a Region Of Practical Equivalence (ROPE) approach, to quantify the chance that an effect is present or absent. Estimates within the ROPE range are considered equivalent to 0 (no effect/difference), whereas estimates outside the ROPE range are considered to have an effect. The estimates are standardised mean differences from the reference category on the latent scale $\tilde{Y}$. Since these estimates are standardised, we used a ROPE range from -0.1 to 0.1 [79].

The analysis was conducted using Stan [80] called via the package brms [81] within the R environment [82]. Furthermore, we used [83–93] for all analyses and reporting. We compared the 95% credible intervals (CrIs) of each estimate and checked whether the CrI fell into the ROPE range. Note: while credible intervals are different from frequentist confidence intervals, the latter often incorrectly get interpreted as the former [74]. If the full CrI fell into the ROPE range, we accepted the null value (there is no effect/difference). If the CrI fell entirely outside the range, we rejected the null value (there is an effect/difference). If it overlapped, we could not make a decision given our data. To give the data more weight than our own assumptions, we applied only weakly informative priors on our parameter estimates: all intercepts/cutoffs and all parameter slopes $\beta$ used a *Normal*(0, 4) prior.

## Results study 1

This section presents results for H1a/b/c and H2a/b/c, which are based on our main analysis of the participants' *intuitive response*. Subsequently, we briefly report the results of our exploratory analysis, using the participants' *informed response*.

## Identity badge versus no badge (H1)

Our first hypotheses (H1a/b/c) claimed that Tweets with an identity verification badge would yield the same sharing intentions and perceived source and message credibility as Tweets without a badge. The output of the analyses can be found in Table 1. Because interpreting main effects in the presence of an interaction can be misleading, we looked at both the main (averaged) effect and each context individually.

The intuitive viewing yielded inconclusive results regarding the effects of identity badges on sharing likelihood. Next, it remained inconclusive whether source credibility is affected by

**Table 1. The effects of an author with an identity badge versus no badge, both for the intuitive and informed viewing.** Results are presented averaged over contexts and for both contexts individually.

| | | Intuitive response | | | Informed response | | |
|---|---|---|---|---|---|---|---|
| | Context | Estimate | SD | 95% CrI | Estimate | SD | 95% CrI |
| **Source Credibility** | **Average** | **0.67** | **0.21** | **[0.26, 1.10]\*** | **1.75** | **0.24** | **[1.29, 2.22]\*** |
| | Medical | 0.97 | 0.30 | [0.38, 1.58]* | 2.04 | 0.32 | [1.43, 2.69]* |
| | Household | 0.37 | 0.30 | [-0.20, 0.96] | 1.45 | 0.32 | [0.84, 2.09]* |
| **Message Credibility** | **Average** | **0.35** | **0.20** | **[-0.03, 0.75]** | **1.26** | **0.20** | **[0.85, 1.66]\*** |
| | Medical | 0.73 | 0.28 | [0.20, 1.28]* | 1.65 | 0.28 | [1.10, 2.19]* |
| | Household | -0.03 | 0.29 | [-0.54, 0.58] | 0.86 | 0.29 | [0.28, 1.43]* |
| **Sharing Likelihood** | **Average** | **0.08** | **0.13** | **[-0.35, 0.51]** | **0.71** | **0.22** | **[0.28, 1.15]\*** |
| | Medical | 0.53 | 0.30 | [0.05, 1.10] | 1.11 | 0.31 | [0.51, 1.72]* |
| | Household | -0.38 | 0.32 | [-1.00, 0.24] | 0.31 | 0.31 | [-0.29, 0.92] |

This table shows the contrasts between the identity badge condition and no badge condition for both credibility measures and sharing likelihood. First, we observed the 95% credible intervals (CrIs) of each estimate and checked whether the CrI fell into the ROPE range. If the full CrI fell into the ROPE range (ranging from -0.1 to 0.1), there is an effect. If the CrI fell entirely outside the range, there is no effect. Present effects are marked with an asterisk (*). If the CrI and ROPE overlapped, we cannot draw a conclusion about the presence or absence of an effect given our data. The estimate and its standard deviation (SD) indicate how the contrasts are distributed.

an identity badge in the household context. However, for the medical context and the average of these contexts, the identity badge substantially increased source credibility. Lastly, message credibility only conclusively increased in the medical context. The household context and the average of the contexts yielded no conclusive results on message credibility. None of these findings support H1a/b/c.

Next, we explored how information about badge interpretation affected user behaviour. Somewhat different results were obtained for this informed viewing, as now identity badges increased source and message credibility for both contexts as well as their averages. Sharing intentions also increased, yet only in the medical context and for the average of the contexts.

**Credential badge vs. no badge (H2).** The credential badges were expected to help people make better credibility judgements. Hence, our second set of hypotheses stated verified credential increases credibility and sharing intentions, but only if the verified credential is relevant to the context (H2a/b/c). We thus compare the effects of a medical credential badge in the relevant (medical) against the irrelevant (household) context. Like the identity badges, the novel credential badges are evaluated with respect to sharing intentions and perceived source and message credibility. The results of the analysis are displayed in Table 2.

**Table 2. The effects of an author with a medical credential badge versus no badge, both for the intuitive and informed viewing.**

| | | Intuitive response | | | Informed response | | |
|---|---|---|---|---|---|---|---|
| | Context | Estimate | SD | 95% CrI | Estimate | SD | 95% CrI |
| **Source Credibility** | Medical | 1.58 | 0.29 | [1.03, 2.17]* | 3.51 | 0.33 | [2.87, 4.18]* |
| | Household | 1.16 | 0.27 | [0.62, 1.70]* | 2.99 | 0.32 | [2.37, 3.64]* |
| **Message Credibility** | Medical | 1.00 | 0.27 | [0.48, 1.51]* | 3.01 | 0.29 | [2.44, 3.58]* |
| | Household | 0.68 | 0.27 | [0.14, 1.21]* | 2.66 | 0.29 | [1.54, 2.66]* |
| **Sharing Likelihood** | Medical | 0.78 | 0.27 | [0.26, 1.31]* | 1.86 | 0.29 | [1.31, 2.42]* |
| | Household | 0.15 | 0.29 | [-0.41, 0.71] | 1.21 | 0.29 | [0.65, 1.79]* |

* = 95% CrI completely outside of the ROPE.

When the credential badge was context-relevant, the badge increased sharing intentions as well as source and message credibility in the intuitive viewing. Yet, when it was irrelevant to the context, the results show the presence of a virtual lab coat effect, as both source and message credibility was inflated by the medical credential badge regardless of the context. As a credential verification badge does not *only* increase source credibility when used in the relevant context, we cannot reject the null hypotheses of H2a/b. Whether this virtual lab coat effect applies to sharing likelihood remains uncertain, leaving us unable to reject the null hypothesis of H2c. Still, for the informed viewing, the credential badge increased all measurements. The exploratory analysis thus evidently shows a virtual lab coat effect.

## Discussion study 1

Study 1 examined the impact of identity and credential verification badges on source credibility, message credibility, and sharing intentions. It was investigated whether identity badges affect user perceptions, expecting no effects (H1). Contrary to prior research [39, 42, 94], identity badges increased source and message credibility for medical posts in the initial viewing, suggesting a conflation of source authenticity and message credibility. These findings on identity verification are somewhat worrisome in the light of misinformation, especially given that identity verification is being made more accessible (e.g., through a subscription [1, 41]). In the household context, however, the effects of identity badges were either inconclusive or much weaker. This may be explained by that household information could be considered as less exclusive compared to medical information. In that case, users rely less on source credibility heuristics [44].

Next, after explaining the interpretation of the badges, medical posts with identity badges yielded higher sharing intentions and perceived source and message credibility compared to posts without a badge. One explanation for the informed viewing findings is that the badges may have triggered users that saw no badge to become more suspicious of deceptive content or sources, leaving their initial assumption of truthful communication [95].

More conclusive findings were observed for credential badges, often larger in effect size compared to the effects of identity badges. Credential badges were expected to increase credibility perceptions and sharing intentions, but only if the verified credential was relevant to the message contents (H2). However, in the intuitive viewing, any posts by sources with verified medical credentials were perceived as more credible and more likely to be shared, even if the credential was irrelevant to the post. Therefore, we evidently show the presence of a "virtual lab coat effect". This suggests that a medical credential lends general credibility, raising concerns regarding the potential of credential verification badges to boost misinformation. The same, even stronger effects were observed for the informed viewing.

Overall, the results show that source credibility increased when authors had identity or credential verification badges, regardless of the relevance of present verified credentials. This expands on research by Alhayan et al. [69], whose participants mentioned they found a source more credible if they had topic-relevant experience or a blue badge. Moreover, this confirms previous findings that any additional information about an online author affects source credibility, as previously researched for online reviews [71–73]. Moreover, we show that source credibility cues affect message credibility, contradicting previous research suggesting that source information has no relationship with the perceived accuracy of online news [96].

One limitation to this experiment is that the post in the medical context contained inaccurate information (see [97]). Though we do not expect participants to have been aware of this, this breaks the underlying assumption that experts generally communicate accurate information. Next, the source credibility ratings might have biased the message credibility ratings, as

they were posed in this order in the survey. Similarly, posing the media literacy and trust questions prior to the intuitive viewing could have biased the participants, raising suspicion of possible deception (e.g., [95]). These limitations will be addressed in study 2.

## Study 2

The second study aimed to reproduce the results of study 1. Another goal of this second study was to verify whether the observed effects of source credibility cues on Twitter generalise to a more generic social media setting. Namely, many other social media platforms also use identity verification badge, e.g., Instagram [40], Facebook [41], and YouTube [98]. These platforms are also obliged to introduce tools that empower users to more accurately determine the credibility of content [32]. It is worth noting that this experiment took place in December 2022, after the upheaval regarding Twitter's identity-based verification badges subscription policy (e.g., [1]).

Though study 2 aimed to test the robustness of the findings in study 1, the results of the first study imply that both badge designs can stimulate reasoning errors with possibly dangerous implications towards misinformation. Therefore, study 2 also introduced a new, different type of authenticity marker: the credential *signature*. This signature communicates the same information as credential verification badge, with one exception: the signature indicates expertise rather than employment. Hence, the author's role within the organisation cannot be disputed, which was a limitation of study 1. An example of a generic social media post with a signature can be found in S2 Fig.

Though the information conveyed by the signature is fairly similar, the design of social media posts can affect user's credibility perceptions (e.g., [12, 37]). The credential signature is expected to reduce the emergence of virtual lab coat effects compared to its badge counterpart. Namely, the signature is placed below the message and therefore likely to be read after the message. This way, the signature information is more likely to be considered in the context of the message, as the context is now already apparent. Verified credentials irrelevant to the message content could thus trigger some form of *expectancy violation* heuristic [52], where messages with an irrelevant signature should not increase the source and content credibility.

In sum, study 2 investigated how identity and credential verification badges generalise to other platforms and compare to credential signatures. As with study 1, the designs are compared on their effects on credibility judgements and sharing intentions. Moreover, the experiment started with a round of intuitive evaluations, followed by explanation on how to interpret the badges and signatures. Consequently, we explored how this explanation affected the credibility and sharing judgements. Based on results of our previous experiment, we hypothesised the following:

- **Hypotheses 3a/b/c**: People are more inclined to find social media posts with an identity verification badge more credible (a: source credibility, b: message credibility) and to share these posts (c: likelihood of sharing), compared to posts without a badge.

- **Hypotheses 4a/b/c**: People are more inclined to find social media posts with a credential verification badge more credible (a: source credibility, b: message credibility) and to share these posts (c: likelihood of sharing), compared to posts without a badge, also if the credential is irrelevant to the context.

- **Hypotheses 5a/b/c**: People are more inclined to find social media posts with social media posts with a credential signature more credible (a: source credibility, b: message credibility) and to share these posts (c: likelihood of sharing), compared to posts without a badge, but only if the credential is relevant to the context.

## Method study 2

The methodology used in our second study replicated exactly that of study 1, with exception for of the details listed below. Before running the experiment, we preregistered the sample size estimation, hypotheses, and planned statistical analyses. The preregistration and all used stimuli, the survey, data, and analysis scripts are available on the Open Science Framework (https://osf.io/y2uew/).

The second study was also approved by the Ethics Committee of Social Sciences at Radboud University. The study was approved as part of the light track (i.e., involving minimal risk). The light track ethical approval used reference number ECSW-LT-2022–12-12-24077. Lastly, as with study 1, participants expressed informed consent through an electronic form prior to the experiment.

**Design.**   The study used a 2x5 between-subjects design. For the first independent variable (social media post context), the household context was replaced with a cybersecurity context, to represent a more exclusive topic and more clearly signal irrelevance of medical expertise. The second independent variable, source credibility cue, now contained five conditions: no badge or signature (control condition), an identity-based verification badge, an credential-based badge showcasing a medical credential, and a credential-based signature showcasing medical expertise. Finally, a credential-based signature showcasing cybersecurity expertise was included to check whether this yields similar effects compared to the medical signature. As this was indeed the case, it will not be reported any further.

**Stimuli and setup.**   As mentioned, the post used for the medical context in study 1 contained false information (see [97]). This was changed to accurate information: the medical context for study 2 thus read "Screening for asymptomatic atrial fibrillation increases the chance of preventing a stroke in elderly patients", based on medical literature [99]. Next, the cybersecurity context media post used a similar framing, and said "The use of connected vehicular cloud computing increases the chance of successful cyber attacks in connected cars", based on cybersecurity literature [100]. An overview of the social media posts in the medical context are displayed in S3 Fig (the other stimuli are available on Open Science Framework).

**Participants.**   On 16 December 2022, we recruited a total of $N = 590$ participants (based on a G*power analysis) through Prolific, using identical sampling criteria to the first study (an age between 18 and 65; current living area in the UK). Participants were compensated with £1.2 for successfully completing the study, which was estimated to take around 8 minutes (£9.00/h). Similar to study 1, a pilot study with 26 people was conducted prior to the main data collection.

Due to sudden technical issues with the server hosting the survey, the survey took participants on average longer than expected ($M = 11.29$, $SD = 9.32$ minutes). The participants received an additional £1.05 to compensate for the issues. We performed an exploratory analysis on the subset of participants who finished the survey within 10 minutes, i.e., likely without technical issues ($N = 297$). In almost all cases, this analysis yielded similar results and effect sizes compared to our main analysis, suggesting the technical difficulties did not affect our results. We thus only report our main analysis.

The sample of 590 participants contained 34 participants with unusable data, 44 participants who did not pass the attention checks. Of these 44, 34 did not remember the contents of the social media post correctly. The remaining 10 thought they recognised the fictional author of the social media account. Hence, they were excluded from analysis. As stated in the pre-registration, we excluded all participants that took over 25 minutes to complete the survey (22 in total), leaving the data of 491 participants for analysis. This population had a mean age of 39.65 ($SD = 11.43$).

**Procedure.** In study 1, the background information questions also contained items on trust in medical organisations and media literacy. Yet, these items were postponed to the end of the survey to prevent accidental priming of expecting deception (e.g., [95]). Similarly, study 2 addressed message credibility prior to source credibility, to circumvent the participants considering the credibility of the message while judging the credibility of the author.

Next, most participants did not correctly remember which badge they had seen in study 1 (62.0%), which is subject to noise as many participants may have guessed. Hence, the answer options to this question were changed by visually supporting the participant, with the aim of reducing noise due to guessing. Namely, instead of displaying an example of the badge, we showed all possible social media posts (using the context the participant was assigned to). Also, the participants were asked to indicate whether they had guessed which post they saw. Now, most participants correctly recognised which post they had seen (72.9%), and the remaining group either guessed or incorrectly recalled. Moreover, this attention check was also included after the second time the participants saw the social media post. For both attention checks, we performed exploratory analyses including only the participants who correctly recalled the source credibility cue. These analyses yielded very similar results to the main analysis in almost all cases. The same holds for the exploratory analyses including the remaining participants. Therefore, they will not be discussed in more detail.

Lastly, the explanation on the designs tested in the survey and their meaning was revised. Specifically, it no longer referred to Twitter, and the text was supported with additional imagery including example posts rather than just badges. The full explanation can be found in S2 File.

**Measures.** Study 2 used the same three background information measures. However, instead of just Twitter use, social media use was assessed using two 10-point Likert-scales, where 1 means *Never* and 10 *Very often* (based on Kim et al. [101]). The two statements addressed how often the participant uses social networking sites (e.g., Facebook) and microblogging websites (e.g., Twitter) to get news. For the analyses, we used the average of these two measures ($r = .32$, $M = 6.30$, $SD = 2.61$). For medical trust, we used the same statement and answer options, which yielded slightly lower results than study 1 ($M = 2.79$, $SD = 0.67$). Media literacy was measured using the same scale [75], also yielding slightly lower results than in study 1 ($M = 5.29$, $SD = 0.85$), $\alpha = .81$.

The sharing likelihood scale was increased to a 7-point Likert-scale aiming to increase granularity and consistency with the other outcome levels. Moreover, the statement was reformulated to "*Imagine that you were with a couple of acquaintances last week. You had a conversation about the topic of the social media post you just saw. How likely are you to share the information in the social media post with them?*", creating a more explicit context for sharing the post. This framing also includes in-person sharing of information, possibly yielding more granularity in results as people tend not to re-share posts on social media (e.g., [56, 58]).

The message credibility scale was revised to resemble the scale by Appelman & Sundar [64] more accurately. Namely, participants now rated the message using the adverbs 'accurate', 'authentic' and 'believable', using a 7-point Likert scale ranging from *Describes very poorly* (1) to *Describes very well* (7). This scale showed very good internal consistency (raw $\alpha = .90$).

Lastly, source credibility was also measured with a 7-point Likert-scale, both for survey consistency and to stick closer to the original source of the source credibility scale [102]. This scale scored good internal consistency (raw $\alpha = .82$). As in study 1, though dropping the item 'biased' would increase $\alpha$ by .11, it was kept for the same reasons.

**Table 3. The effects of an author with an identity badge versus no badge, both for the intuitive and informed viewing.** Results are presented averaged over contexts and for both contexts individually.

| | | Intuitive response | | | Informed response | | |
|---|---|---|---|---|---|---|---|
| | **Context** | **Estimate** | **SD** | **95% CrI** | **Estimate** | **SD** | **95% CrI** |
| **Source credibility** | **Average** | **-0.06** | **0.27** | **[-0.58, 0.45]** | **1.36** | **0.28** | **[0.81, 1.89]**\* |
| | Medical | 0.21 | 0.39 | [-0.53, 0.96] | 1.68 | 0.40 | [0.90, 2.46]\* |
| | Cyber security | -0.34 | 0.36 | [-1.04, 0.39] | 1.03 | 0.36 | [0.33, 1.73]\* |
| **Message credibility** | **Average** | **-0.22** | **0.27** | **[-0.73, 0.30]** | **1.17** | **0.28** | **[0.64, 1.72]**\* |
| | Medical | -0.24 | 0.39 | [-1.02, 0.52] | 1.13 | 0.40 | [0.37, 1.93]\* |
| | Cyber security | -0.20 | 0.36 | [-0.90, 0.52] | 1.21 | 0.37 | [0.51, 1.94]\* |
| **Sharing likelihood** | **Average** | **-0.12** | **0.27** | **[-0.65, 0.42]** | **0.55** | **0.26** | **[0.05, 1.08]** |
| | Medical | -0.31 | 0.39 | [-1.08, 0.45] | 0.47 | 0.38 | [-0.27, 1.24] |
| | Cyber security | 0.07 | 0.36 | [-0.77, 0.63] | 0.64 | 0.35 | [0.04, 1.34] |

\* = 95% CrI completely outside of the ROPE.

## Results study 2

This section presents the results for H3 to H5. As with the first study, the main analysis was based on the participants' *intuitive response*. The informed responses, i.e., ratings after the second viewing where the participant had information about the various badges, are reported as exploratory analysis.

**Identity badge versus no badge (H3).** As we aimed to replicate the results of study 1 in a more general context, our third hypothesis claimed that people are more inclined to share social media posts with an identity verification badge, and to find these posts and their source more credible compared to posts without a badge. The posterior distributions with mean, standard-deviation and 95% CrI of the difference between the identity badge and no badge for the three outcome variables are listed in Table 3.

For the intuitive viewing, it remained inconclusive whether identity badges increase sharing intentions or credibility judgements. The null hypotheses of H3a/b/c can thus not be rejected, as no effects could be detected in both contexts. Additionally, we explored how sharing and credibility judgements were affected by an explanation on the interpretation of the badges and signature. For this informed viewing, it also remained inconclusive whether identity badges increase sharing intentions. In contrast, source and message credibility increased in both contexts as well as their average. The account holder and their message were thus thought to be more credible after the explanation, showing a conflation between source authenticity and message credibility.

**Credential badge versus no badge (H4).** To investigate our fourth set of hypotheses, we compared social media posts with an credential verification badge to posts without a badge in terms of their effects on sharing intentions and source and message credibility. It was hypothesised that these measurements increase for posts with an credential verification badge, regardless of whether the credential is relevant to the context. For the analysis, we considered the medical credential verification badge. The results of our analysis are listed in Table 4.

For the intuitive viewing, a relevant credential badge increased the credibility of a source. However, it remains inconclusive how source credibility is affected when the badge is context irrelevant. Moreover, the effects of a credential badge on sharing likelihood and message credibility also remain inconclusive. Hence, the null hypotheses of H4a/b/c could not confidently be rejected, as the data does neither illustrate an increase in the relevant context nor the

**Table 4. The effects of an author with a medical credential badge versus no badge, both for the intuitive and informed viewing.**

| | Context | Intuitive response | | | Informed response | | |
|---|---|---|---|---|---|---|---|
| | | Estimate | SD | 95% CrI | Estimate | SD | 95% CrI |
| **Source Credibility** | Medical | 0.80 | 0.36 | [0.13, 1.52]* | 2.46 | 0.38 | [1.71, 3.19]* |
| | Cybersecurity | 0.01 | 0.35 | [-0.67, 0.69] | 1.51 | 0.35 | [0.84, 2.18]* |
| **Message Credibility** | Medical | 0.09 | 0.37 | [-0.62, 0.81] | 2.01 | 0.38 | [1.29, 2.75]* |
| | Cybersecurity | 0.24 | 0.34 | [-0.42, 0.91] | 1.62 | 0.36 | [0.93, 2.33]* |
| **Sharing Likelihood** | Medical | -0.27 | 0.36 | [-0.97, 0.42] | 1.05 | 0.37 | [0.34, 1.78]* |
| | Cybersecurity | -0.13 | 0.35 | [-0.80, 0.53] | 0.75 | 0.34 | [0.08, 1.43] |

\* = 95% CrI completely outside of the ROPE.

absence of a virtual lab coat effect. Next, for the informed ratings, a relevant credential badge yielded increased sharing intentions. However, it remains unclear whether this effect also occurs in the context irrelevant setting. Yet, the credential badge increased the informed source and message credibility ratings, regardless of its relevance to the message content.

**Credential signature versus no signature (H5).**  The final set of hypotheses claimed that posts with a signature increase sharing intentions and source and message credibility judgements compared to posts without a signature, but only if this credential information is relevant to the context. The parameters of the posterior distributions are listed in Table 5.

Similar to the credential badge, the credential signature yields an increase in source credibility when the verified credential is context relevant. Yet, it remains unclear whether this effect on source credibility also occurs when the credential is context irrelevant. Moreover, the data yields no definitive conclusions on the effects of credential signature on sharing likelihood and message credibility. In sum, none of null hypotheses of H5a/b/c can reliably be rejected. Lastly, for the informed ratings, a source with a credential signature was perceived as more credible regardless of the relevance of the credential within the context. Next, relevant credential signatures increased sharing intentions and message credibility. Yet, it remains inconclusive whether this also holds for credential signatures in irrelevant contexts.

## Discussion study 2

The goal of study 2 was to test whether the results of study 1 generalise to a broader social media context. Moreover, it introduced a credential-based signature, as a potential solution to the virtual lab coat effects of the credential-based verification badge observed in study 1. In

**Table 5. The effects of an author with a medical credential signature versus no signature, both for the intuitive and informed viewing.**

| | Context | Intuitive response | | | Informed response | | |
|---|---|---|---|---|---|---|---|
| | | Estimate | SD | 95% CrI | Estimate | SD | 95% CrI |
| **Source Credibility** | Medical | 1.44 | 0.37 | [0.71, 2.19]* | 2.29 | 0.39 | [1.53, 3.06]* |
| | Cybersecurity | 0.10 | 0.35 | [-0.58, 0.82] | 1.15 | 0.36 | [0.43, 1.84]* |
| **Message Credibility** | Medical | 0.61 | 0.36 | [-0.08, 1.35] | 2.00 | 0.38 | [1.27, 2.72]* |
| | Cybersecurity | -0.32 | 0.35 | [-1.00, 0.37] | 0.52 | 0.37 | [-0.22, 1.23] |
| **Sharing Likelihood** | Medical | 0.08 | 0.38 | [-0.68, 0.82] | 0.95 | 0.38 | [0.24, 1.74]* |
| | Cybersecurity | -0.79 | 0.36 | [-1.50, -0.07] | -0.34 | 0.36 | [-1.04, 0.38] |

\* = 95% CrI completely outside of the ROPE.

contrast to study 1, it remained inconclusive whether there exists an intuitive relationship between verified identity badges and source and message credibility, or sharing intentions. Therefore, H3 was not supported. However, the informed viewing data provides more support for the conflation of source authenticity and message credibility. Namely, after the explanation on how to interpret the badges and signature, participants rated posts by unknown authors with a verified identity as more credible, suggesting an incorrect reasoning shortcut. Note, however, that the difference in credibility and sharing intentions after explanation often stemmed from lower ratings for posts from authors without any verified information. This holds for all contrasts.

The hypotheses on credential-based badges (H4) were also not supported, contrary to results of study 1. It was expected that credential-based badges increased perceived source and message credibility as well as sharing intentions, regardless of the whether the credential information was context relevant. Yet, the credential-based verification badge only conclusively increased source credibility when relevant credential information was verified. It thus remains inconclusive whether a virtual lab coat effect occurs for the intuitive viewing. However, after explaining the interpretation of the badges and signature, the virtual lab coat effect was evident. Even if an unknown author's verified credential information badge was irrelevant to the context, their message was perceived as more credible. Hence, although the effect size is much larger in the relevant context, a verified medical credential badge still lends credibility to irrelevant contexts.

The credential-based signatures were expected to increase perceived source and message credibility as well as sharing intentions. Yet, only when the credential information was context relevant (H5). The effects of the credential-based signature were very similar to its badge counterpart. However, there are two notable differences. First, the effect size for source credibility during the intuitive viewing was much higher for relevant signatures than for relevant badges. Second, it remains inconclusive whether the informed viewing yields a virtual lab coat effect. An explanation for this is the novelty of the design, which might have led participants to process the signature more elaborately (e.g., [103]).

The similarity between credential-based signatures and badges contradicts previous studies that suggest source cues are less impactful when they appear after the message [104, 105]. Yet, those studies included participants motivated to think critically, which is not necessarily the case in this study. Moreover, the similarity in results between credential-based badges and signatures is particularly interesting, as the signatures conveyed more information than their badge counterpart (ensuring the author works as medical expert instead of, e.g., administrator). Hence, though this was listed as a limitation in study 1, it seems unlikely it affected participants' reasoning. On the other hand, the more explicit medical expertise could explain the larger effect size of the signature compared to the badge variant in the intuitive viewing.

The results of study 2 provide less support for our previous finding that verified information about an unknown source increases their credibility. Namely, though this notion held for the informed viewing, the intuitive viewing only conclusively yielded higher author credibility in case of relevant verified credential information. For identity-based verification or irrelevant credential information, no definitive conclusions could be drawn about their relationship with source credibility.

Notably, study 1 and study 2 provide somewhat different results, especially for the intuitive viewing. This discrepancy can likely be attributed to the more generic style the posts were presented in. Alternative explanations are small changes in methodology, and possible associations with the turbulence regarding Twitter's identity-based verification badges around the time the experiment was conducted (e.g., [1]).

## General discussion

To counter online misinformation, this paper proposed to empower users through online platforms with tools to determine the credibility of social media posts more accurately. Specifically, we suggested to focus on source credibility cues, as users heavily assess the credibility of content through the lens of source credibility [35]. As source information, social media platforms commonly employ an identity verification badge, verifying an authors identity. In addition to such verification badges, this paper investigated the novel possibility of credential verification on social media. Though verifying credentials and identity are both technically viable [2], it was unclear how they affected credibility judgements and sharing behaviour. This was investigated through two experimental studies ($N = 525; N = 590$), where participants rated a social media post with respect to credibility and sharing likelihood. Participants rated a post both before and after explanation of how to interpret the verified source information cues.

Before explanation, participant behaviour was unpredictable in a general social media setting. Namely, in most cases, both identity and credential verification methods did not show a clear absence or presence of effects on sharing intentions, and source or message credibility. Our results thus question the omnipresence of verified identity markers on social media platforms, as they do not seem to increase source credibility intuitively. However, specifically in an environment simulating Twitter, medical messages from unknown sources seemed more credible if the source had verified their identity or medical credential. Whereas a verified medical credential is a useful heuristic for judging the credibility of medical messages, a verified identity absolutely is not. Inflated medical message credibility in case of a verified identity is a worrisome finding, given that users increasingly use social media for medical information [7, 8], and that verifying an identity is publicly available as a subscription [1]. This result likely applies to other messages of which the user has no intuition about its credibility [44].

However, most reasoning errors occurred after the participants were informed about how to interpret the verified source information cues. In both the Twitter setting as the general social media setting, participants often rated posts with verified source information cues as substantially more credible compared to posts without these cues after this explanation. The results surfaced two prominent reasoning errors in participants. First, participants conflated source authenticity and message credibility. Second, participants often found messages from sources with a verified credential more credible, even if this credential was irrelevant to the message. It thus seems that verifying information about an unknown source can increase the source's credibility, as also suggested by previous studies [71–73]. Generally, verified source information is a good heuristic for source credibility. Yet, it becomes problematic when irrelevant verified source information (e.g., their identity, or irrelevant credential) inflates the credibility of the message. In its currently presented forms, verified source information thus seems to be more able to boost misinformation rather than to hinder it.

As most reasoning errors occurred after informing participants about the verified source information cues and what they mean, purely explaining how to interpret them seems insufficient to overcome the prevalence of these reasoning errors. In other words, this simplest form of a (news) media literacy intervention did not have the anticipated effect. This calls into question to what extent such low profile interventions are useful in combating misinformation, and subsequently how media literacy interventions should look like to be effective. Past research (e.g., [106, 107]) indicates that more sophisticated interventions (e.g., integrating several strategies and/or featuring multiple messages) are promising. However, it is crucial to note that our theoretical suggestions are based on exploratory research, and we refrain from making definitive conclusions. Further investigation is required, particularly when it comes to

the existence of negative side-effects of exposure to interventions [107], as also found in the current study.

Lastly, both studies are mostly inconclusive on the relationship between verified source information and intentions to share their posts. A likely explanation for this finding is that credibility and sharing are not necessarily related (e.g., [29, 56, 57]), and that sharing intentions have been found to depend on factors uncontrolled for, e.g., someone's motivation to protect their self-image [60], or the novelty of the information [54]. Furthermore, it is hard to detect effects on sharing, as only few people tend to often re-share information by others on social media (e.g., [56, 58]).

## Limitations and future work

Some caveats to these results are to be noted. First, to minimise cultural noise, our participants were all based in the UK. However, cultural differences can also affect credibility processing (e.g., [38]). Moreover, some participants might have had prior knowledge about the social media posts contents. Though prior knowledge was assumed to be rare and distributed equally among conditions, it might still have affected their credibility judgements (e.g., [44]). Hence, future research could investigate how verified source information affects user behaviour for users from a wider variety of cultural and educational backgrounds.

This paper varied the presentation of (verified) source information and its effects on user behaviour, and the relevance of these cues to the message content (i.e. relevant or irrelevant expertise). However, another interesting experimental variation would be to vary information accuracy within the experiments. Using such a ground-truth is common practice in misinformation research. We purposely left this variation out considering that the 'truth' is often emergent and subjective, but future work could certainly experiment with more objective truths or falsehoods.

Furthermore, the stimuli contained rather neutral, scientific topics rather than polarising topics. This circumvents potential motivated reasoning where users pay less attention to source cues and more to message contents [108]. Still, sources affect message credibility as, e.g., messages from sources aligning with users' political bias are perceived as more credible [109]. Yet, further research is needed to understand how verified source information of unknown sources affects content evaluation in polarising contexts. Additionally, it must be noted that scientific content can increase source credibility [69]. Therefore, another interesting angle for future research is how less scientific content (e.g., opinions, experiences) are evaluated in case verified source information is present.

Next, a potential downside of verifying credentials is its implication on anonymity. By verifying additional information, one's personal data is shared on the internet. Users might be inclined to disclose more data (than they might have self-disclosed in their account biography) to gain credibility, without considering the impact on their privacy. Therefore, although verified credentials could improve credibility judgements in theory, the observed virtual lab coat effects and privacy risks associated with this technology underline it is inadequate to deploy it on social media in the form featured in this paper. Still, verified credentials could be employed in contexts less cognitively demanding than social media. An example of such a context where source credibility is important is Wikipedia (see, e.g., [110]).

Lastly, the experimental survey presented the sharing and credibility items all in the same fixed order. Though this order was fixed to circumvent specific priming effects (sharing intentions could be primed by credibility judgements), other priming effects may have occurred (e.g., sharing intentions priming credibility judgements for internal consistency). Therefore,

the results of our study would be more robust if the survey included a randomised presentation order of these items instead.

## Conclusion

With the introduction of social media, platforms hardly have any control over what users can or cannot say (e.g., [5]), making way for misinformation. Yet, determining the credibility of messages can be troublesome for users (e.g., [10]). Therefore, social media platforms should strive to make users smarter in what they believe. A good starting point is a more accurate perception of the credibility of the message's source (e.g., [36]). We have investigated how verified source information influences credibility judgements and sharing intentions. We examined (1) an existing implementation, i.e., verified source identity, and (2) a potential new solution, i.e., verified credentials of the source.

Without providing users with knowledge on how to interpret the verified source information cues, the effects of these cues were mostly unpredictable. Yet, after explanation, users were prone to two reasoning errors. First, they conflated source authenticity and message credibility. Second, messages from sources with a verified credentials were more credible, regardless of whether this credentials was context relevant (i.e., virtual lab coat effect). The prevalence of these two reasoning errors seem especially harmful in the light of misinformation. Hence, future research should investigate how different solutions can be more effective in empowering users to more accurately determine the credibility of social media posts. Furthermore, alternative implementations or contexts to implement credentials verification should be explored, as it a promising development in the area of online credibility.

## End note

For full transparency, it must be noted that during revision of the paper, all hypotheses were slightly reworded to clarify the distinction between credibility judgements and sharing intentions, based on that these decisions are not necessarily related (e.g., [29, 56, 62]). This theoretical distinction did not affect how we conducted our analyses. One exception to this is the reformulation of hypothesis 4. Namely, the pre-registered hypothesis stated that credential-based verification badges do not affect the credibility judgements and sharing intentions with respect to posts without a badge. Yet, given the results from study 1, the hypothesis should have stated an increase of these factors.

## Supporting information

**S1 Fig. Medical context stimuli study 1.** An overview of the stimuli used in the medical context in study 1. Here, the medical signature (c) is considered a relevant attribute, as it is displayed in the medical context. Note that this figure is for illustrative purposes only for two reasons. First, the profile photo is similar but not identical to the one used in the experiment. While the original photo was obtained from Unsplash, this illustrative profile picture was AI-generated through https://thispersondoesnotexist.com. Second, this Figure differs from the original stimulus in that it features a self-designed version of a Tweet for legal reasons. (TIF)

**S2 Fig. Credential signature on generic social media post.** Example social media post including a credential signature. Note that this image was used as support image in explaining the meaning of the various badge and signature designs. Note that, again, this figure is for illustrative purposes only, as the profile photo is similar but not identical to the one used in the experiment. While the original photo was obtained from Unsplash, this illustrative profile picture

was AI-generated through https://thispersondoesnotexist.com.
(TIF)

**S3 Fig. Medical context stimuli study 2.** An overview of the stimuli used in the medical context in study 2. Here, the medical signature (d) is considered a relevant credential, whereas the cybersecurity signature (e) is considered irrelevant. Naturally, the opposite holds in case of the cybersecurity context (where only the message contents are replaced). Note that, again, this figure is for illustrative purposes only, as the profile photo is similar but not identical to the one used in the experiment. While the original photo was obtained from Unsplash, this illustrative profile picture was AI-generated through https://thispersondoesnotexist.com.
(TIF)

**S1 File. Explanation of the designs used in study 1.** This text was used to explain to the participants how to interpret the different possible badges included in the first experiment. It was explained that a source could have either no badge, an identity-based badge, or a credential-based badge. The text further explained how to interpret the badges, namely as verified identity and verified credential respectively. Lastly, this File differs from the original stimulus in that it features a self-designed version of the Twitter verification icon for legal reasons.
(PDF)

**S2 File. Explanation of the designs used in study 2.** This text was used to explain to the participants how to interpret the different possible badges included in the second experiment. It was explained that a source could have either no badge, an identity-based badge, a credential-based badge, or signed their message using a credential-based signature. The text further explained how to interpret the badges and signature, namely as verified identity and verified credential respectively. Note that, again, this figure is for illustrative purposes only, as the profile photo is similar but not identical to the one used in the experiment. While the original photo was obtained from Unsplash, this illustrative profile picture was AI-generated through https://thispersondoesnotexist.com.
(PDF)

## Acknowledgments

We would like to thank Lian for helping with creating the stimuli for study 2. Next, we thank Bernard van Gastel, Koen Verdenius, Marie-Sophie Simon, Emma Schipper, and Yelyzaveta Markova for their inspiring work and ideas. We thank Bart Jacobs for discussion and originally posing the idea of using verified credentials to guarantee authenticity of information in the context of misinformation. We thank our reviewers for their thoughtful remarks and suggestions.

Lastly, we note that we have used the Artficial Intelligence tool https://thispersondoesnotexist.com to generate profile pictures of non-existing people in S1–S3 Figs, and S2 File. We regenerated portraits until the model yielded a portrait similar to the profile picture used in the study's stimuli, which we could not include for legal reasons.

## Author Contributions

**Conceptualization:** Jorrit Geels, Paul Graßl, Hanna Schraffenberger, Martin Tanis, Mariska Kleemans.

**Data curation:** Jorrit Geels, Paul Graßl.

**Formal analysis:** Jorrit Geels, Paul Graßl.

**Investigation:** Jorrit Geels, Paul Graßl.

**Methodology:** Jorrit Geels, Paul Graßl, Hanna Schraffenberger, Martin Tanis, Mariska Kleemans.

**Project administration:** Jorrit Geels, Paul Graßl.

**Resources:** Jorrit Geels, Paul Graßl.

**Supervision:** Hanna Schraffenberger, Martin Tanis, Mariska Kleemans.

**Validation:** Jorrit Geels, Paul Graßl.

**Visualization:** Jorrit Geels, Paul Graßl.

**Writing – original draft:** Jorrit Geels.

**Writing – review & editing:** Jorrit Geels, Paul Graßl, Hanna Schraffenberger, Martin Tanis, Mariska Kleemans.

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
