## [Decision Letter · Decision Letter 0]

21 Sep 2023

PONE-D-23-22781Virtual Lab Coats: The Effects of Verified Source Information on Social Media Post CredibilityPLOS ONE

Dear Dr. Geels,

Thank you for submitting your manuscript to PLOS ONE. After careful consideration, we feel that it has merit but does not fully meet PLOS ONE’s publication criteria as it currently stands. Therefore, we invite you to submit a revised version of the manuscript that addresses the points raised during the review process.

ACADEMIC EDITOR: 

Dear authors,

Thank you for submitting your work to PLOS ONE. We have now received a feedback from two external reviewers. As indicated in their comments below, both reviewers recognize the relevance of your work. However, they have also raised several concerns and provided detailed suggestions on how the manuscript could be improved. You should answer all their comments point by point and revise according to them, or clearly state if you have another opinion. 

I am looking forward to receive a revised version of your manuscript.

Nicola Diviani

Academic Editor

We look forward to receiving your revised manuscript.

Kind regards,

Nicola Diviani

Academic Editor

PLOS ONE

4. We note that Figures 1, 2 and 3 in your submission contain copyrighted images. All PLOS content is published under the Creative Commons Attribution License (CC BY 4.0), which means that the manuscript, images, and Supporting Information files will be freely available online, and any third party is permitted to access, download, copy, distribute, and use these materials in any way, even commercially, with proper attribution. For more information, see our copyright guidelines: http://journals.plos.org/plosone/s/licenses-and-copyright.

1. You may seek permission from the original copyright holder of Figures 1, 2 and 3 to publish the content specifically under the CC BY 4.0 license.

Reviewers' comments:

Reviewer's Responses to Questions

**Comments to the Author**

1. Is the manuscript technically sound, and do the data support the conclusions?

Reviewer #1: No

Reviewer #2: Yes

2. Has the statistical analysis been performed appropriately and rigorously? 

Reviewer #1: No

Reviewer #2: I Don't Know

3. Have the authors made all data underlying the findings in their manuscript fully available?

Reviewer #1: Yes

Reviewer #2: No

4. Is the manuscript presented in an intelligible fashion and written in standard English?

Reviewer #1: Yes

Reviewer #2: Yes

5. Review Comments to the Author

Reviewer #1: This study examined the effects of source verification (identity, credential, no verification) and message content (e.g., health vs. household related) on perceived source and message credibility as well as sharing intentions of twitter and social media posts at two different time points (e.g., before and after explaining how to interpret source verification information). Results indicated that messages from sources with a verified credential were perceived as more credible, regardless of whether this credential was content relevant, suggesting that credential verification was ineffective in empowering users. While this study investigated interesting research questions on a timely topic and I liked some experimental approaches adopted in the study (e.g., separating between source and message credibility, keeping non-manipulated constant), I have some concerns about conceptualization of the study and related methodological setup employed in the paper, as detailed below.

1) The authors framed that the paper’s main goal was to empower users toward combating misinformation, and to do this they examined the changes in sharing behavior before and after informing participants on how to interpret source verification cues. Thus, the study rationale was built on the assumption that sharing is related to judgements of accuracy. The recent studies, however, showed that sharing and truth judgement can be independent/unrelated (Pennycook & Rand, 2021 https://doi.org/10.1016/j.tics.2021.02.007). Given this evidence, the rationale behind indexing sharing behavior as a metric for measuring susceptibility to misinformation lacks support from the literature. Also, all the stimuli used in the study were fictional, without proper manipulation of ground truth for posts (e.g., real vs fake). This aspect of the design causes any conclusions made regarding misinformation (or about accuracy of sharing intentions--as the authors wrote in the abstract) from the paper to be erroneous/illogical given there were no ground truth.

2) In the study, participants were asked to report on their likelihood of sharing and before making any credibility judgments, and the authors reasoned that this was done for preventing any bias (from thinking about credibility) when reporting sharing decision. I am not sure this really helped because the participants were already primed about credibility after viewing questions for the first post. I think the proper experimental technique to deal with this problem is to counterbalance presentation order of questions between (or maybe even within) participants.

3) Was there a measure on the medical knowledge level of participants (e.g., some may medical expertise)? This may have influenced some of the results from the study.

4) What was the reason for not including first and second round time point as a factor in the statistical model, but conducting two different statistical models to test effects at these time points instead? This practice may have inflated Type-I errors in results.

Reviewer #2: Thank you for the opportunity to review the manuscript Virtual Lab Coats: The Effects of Verified Source Information on Social Media Post Credibility. The manuscript investigates the effects of different badges and signatures that can inform social media users about the message’s source. As the authors address in the paper, this type of information which helps users/readers to make an estimate about the message accuracy is especially important in the context misinformation, which is currently widespread on social media. What I particularly like about the paper, next to the interesting and important topic, is that they approached the research very thoroughly and in a very transparent way. The manuscript describes two studies. One of the aims of the second study was to replicate the findings of the first. The studies were preregistered and all deviations from the preregistration are clearly described and motivated.

When reading through the manuscript some things raised my mind, which I will share below and hope will be helpful to the authors to further strengthen their manuscript. In general, my points move chronologically through the manuscript, starting with the introduction and ending with the discussion, but here and there I deviate from that.

Major points

1. The introduction and literature review is extensive and much related work is discussed. However, as the manuscript is quite information dense and both the independent variables (types of source verification) and the dependent variables source and message credibility in the study are conceptually quite close to each other, it would help the reader if this section was very clearly structured. In the current setup, the introduction consists of many practical and scientific reasons to conduct this study, but they are not presented in a very clear order. Therefore it took me quite some time to understand what problem the manuscript was exactly addressing and especially what the difference between the different badges was. A more traditional structure of an introduction, in which first the social developments (or problems) are described that the paper aims to solve, followed by the aim and a structured theory section including definitions of key concepts, would prevent the reader (or at least: me) from getting lost. Perhaps some restructuring of this section can also ensure that the background can be shortened a little bit, because I think there was some repetition here and there.

On a related note: For me it would help if the practical problem that arises on Twitter (or X nowadays) was first clearly outlined; not only the problem associated with misinformation (which is currently discussed) but also the problem that the currently applied methods of source verification say nothing about source credibility, or message credibility. That point is made later, but since the conflation of source verification and message credibility is quite central to the manuscript, it might be useful to mention this in the beginning.

2. The background extensively discusses source credibility and message credibility as dependent variables, but sharing intentions are less discussed. The importance of this variable is briefly mentioned from a practical perspective, but less from a theoretical/literature angle. Later in the manuscript, the authors indicate that credibility evaluations and sharing intentions do not necessarily have to be related and that the latter are also influenced by other factors. Nevertheless, these variables are often examined together in studies and it might therefore be good to pay more attention to the (theoretical) background of this variable, even if the variables are less related than expected.

3. On page 6 (line 222) the authors state that they use Bayesian analyzes to obtain more certainty about the relationship between source credibility cues and the measures. What is meant by "more certainty"? And compared to what exactly?

4. The authors state that the hypotheses of study 1 are based on Vaidya's study. Because this study has such an influence on this paper, it would be helpful to the reader to get a little more background about this study. For example, what do the authors provide as the reason that there is no difference in effect on the dependent variable between the messages with and without a badge?

5. Hypothesis 1 a/b/c assumes no effect of the independent variable on the dependent variable. As I am not a statistician I cannot properly judge whether this is a problem or not. I appreciate the extensive analysis section. Yet it is still difficult to understand for readers with limited knowledge of Bayesian models. In particular, I would like to read here how this type of statistic is suitable for testing hypotheses without an effect.

6. A power analysis was performed to determine the number of subjects. Is this also suitable for Bayesian analyses?

7. The results for both studies consist of two parts; outcomes for the intuitive response and for the informed response. Ultimately, the results also show that there are considerable differences between them. This shows that the 'explanation about the interpretation of the badges' is crucial, especially because the authors ultimately believe that it does more harm than good. That is why, as a reader, I would like to see the exact explanation that the participants received, in the text, or as a table or figure. This would be informative. In addition, it might be good if the authors delved a little deeper into the theoretical implications of this finding in the discussion. Now this mainly happens on a practical level - for example by finding that simple media literacy interventions may not work sufficiently.

8. To be honest, I found it hard to interpret the tables. Presenting the dependent variable on the far left column is a bit counterintuitive. But I had the greatest difficulty with the estimate that was presented. What does this entail? Is this a difference score between the average score with badge and without badge? For me it would be much more informative if the mean scores on each dependent variable were presented for the different types of badges (and no badge). The difference scores (or effect sizes) that are now in the table could be included in the text when the authors discuss a specific effect.

9. It might be good to provide some reflection in the discussion section on the choice of testing these effects with false or accurate information provided in the messages. This varied across the two studies if I understood correctly (study 1 false information, study 2 accurate information) but could this potentially have impacted people's evaluation? What is the role of prior knowledge here?

10. The authors indicated that the data of the studies are publicly available, but for me it was unclear how to access the data. This could be a problem with the system, however, it means that I could not review the data.

Minor points

1. The authors state that a generic name and profile photo have been chosen for the Tweets, from the appendix I deduce that the sender is a woman. Was there a consideration behind this?

2. I think the measures section could be a little more structured. Now the concepts are hidden in the long paragraph. A paragraph per measure is clearer.

3. The authors use the media literacy scale of Vrage et al. Is this a validated scale? And can the authors provide some sample items? The description of this measure refers to item 6, but without further representation of the scale, 'item 6' has no meaning for the reader. Better to give the substantive item.

4. Why was trust in organizations not added as a control variable and twitter use and media literacy included?

5. Page 2, line 10: this states that the paper examines 3 ways to counter misinformation on social media, but more precisely: ways to counter the impact/effects of misinformation.

6. Other wording issue: page 18, line 753 states that for the intuitive viewing participant behavior was unpredictable. I would advise the authors to be more precise here, for example that in this case the source credibility cue did not impact message evaluation.

7. The term “truth evaluations/assessment” appears in various places in the manuscript. For example on page 2, line 22. The author may consider replacing this with accuracy evaluations.

8. Page 4, line 112; in should be an; Page 5, line 161; represent represent (redundant word). Same applies to page 13, line 520.

9. Page 19, line 792: In my view 'explanation for this finding' better fits the context than 'explanation for this data'.

6. PLOS authors have the option to publish the peer review history of their article (what does this mean?). If published, this will include your full peer review and any attached files.

Reviewer #1: **Yes: **Didem Pehlivanoglu

Reviewer #2: **Yes: **Corine Meppelink

---

## [Author Response · Author response to Decision Letter 0]

28 Feb 2024

Dear Didem, dear Corine,

Thanks for your attention to our manuscript! We elaborately revised our manuscript in accordance with your valuable suggestions and included more clarification where necessary. An overview can be found in the response letter.

Best,

The authors

---

## [Decision Letter · Decision Letter 1]

2 Apr 2024

Virtual Lab Coats: The Effects of Verified Source Information on Social Media Post Credibility

PONE-D-23-22781R1

Dear Dr. Geels,

We’re pleased to inform you that your manuscript has been judged scientifically suitable for publication and will be formally accepted for publication once it meets all outstanding technical requirements.

Kind regards,

Nicola Diviani

Academic Editor

PLOS ONE

Additional Editor Comments (optional):

Reviewers' comments:

Reviewer's Responses to Questions

**Comments to the Author**

1. If the authors have adequately addressed your comments raised in a previous round of review and you feel that this manuscript is now acceptable for publication, you may indicate that here to bypass the “Comments to the Author” section, enter your conflict of interest statement in the “Confidential to Editor” section, and submit your "Accept" recommendation.

Reviewer #2: All comments have been addressed

2. Is the manuscript technically sound, and do the data support the conclusions?

Reviewer #2: Yes

3. Has the statistical analysis been performed appropriately and rigorously? 

Reviewer #2: Yes

4. Have the authors made all data underlying the findings in their manuscript fully available?

Reviewer #2: Yes

5. Is the manuscript presented in an intelligible fashion and written in standard English?

Reviewer #2: Yes

6. Review Comments to the Author

Reviewer #2: (No Response)

7. PLOS authors have the option to publish the peer review history of their article (what does this mean?). If published, this will include your full peer review and any attached files.

Reviewer #2: No

---

## [Editor Report · Acceptance letter]

1 May 2024

PONE-D-23-22781R1 

PLOS ONE

Dear Dr. Geels, 

I'm pleased to inform you that your manuscript has been deemed suitable for publication in PLOS ONE. Congratulations! Your manuscript is now being handed over to our production team.

Kind regards, 

on behalf of

Dr. Nicola Diviani 

Academic Editor

PLOS ONE